# Assessing the Effectiveness of Qista Baited Traps in Capturing Mosquito Vectors of Diseases in the Camargue Region (France) and Investigating Their Diversity

**DOI:** 10.3390/ani13111809

**Published:** 2023-05-30

**Authors:** Mehdi Boucheikhchoukh, Ismail Lafri, Anlamina Chamssidine Combo, Christophe Regalado, César Barthés, Hamza Leulmi

**Affiliations:** 1Department of Veterinary Sciences, Chadli Bendjedid El Tarf University, PB 73, El-Tarf 36000, Algeria; m.boucheikhchoukh@univ-eltarf.dz; 2Department of Veterinary Sciences, Blida 1 University, Blida 09000, Algeria; lafrismail@gmail.com; 3Qista Techno BAM, 130 Lubéron Avenue, 13560 Sénas, France; 4Department of “Licence Sciences et Technologies”, Université Grenoble Alpes, 480 Avenue Centrale Domaine Universitaire, 38400 Saint-Martin-d’Hères, France; 5Department of “Génie Biologique”, Aix-Marseille Université, 19 Boulevard Saint Jean Chrysostome, 04000 Digne les Bains, France; 6Department of “Génie Biologique”, Université de Caen Basse Normandie-Campus 2, Boulevard du Maréchal Juin, CEDEX 5, 14032 Caen, France

**Keywords:** mosquito, BAM, QISTA, Saint-Gilles, France

## Abstract

**Simple Summary:**

Mosquitoes are responsible for causing various disturbances, such as nuisance, allergies, and acting as the vectors of deadly diseases for humans and animals. To combat this confirmed vector, various strategies are implemented. One such strategy is the use of the Qista trap (also known as the Borne-Anti-Moustique: BAM), which employs a CO_2_ and lure emitting mechanism to bait mosquitoes. In this study, six BAMs were deployed to assess their effectiveness in reducing the nuisance rate caused by mosquitoes. The traps were able to capture 84,461 mosquitoes from eleven different species over a period of six months. The average capture rate per trap was 76.92 mosquitoes per day. The survey indicated a significant reduction in nuisance rates in the protected area compared to the control area. The Qista BAM trap appears to be a promising tool for mosquito control and to update the biodiversity of mosquito seeking species in the south of France.

**Abstract:**

Nuisance, allergy, and vector role: mosquitoes are responsible for numerous inconveniences. Several strategies have been employed to fight against this confirmed vector. To record the diversity of mosquito vectors in Camargue (France) and assess the effectiveness of the Qista trap, six BAMs were deployed as a belt barrier to protect the Espeyran Castle (Saint-Gilles, Camargue). Prior to evaluating the reduction in the nuisance rate, recovery nets from the traps and human landing catches (HLC) were utilized twice a week in the treated and control areas. Overall, 85,600 mosquitoes were captured, belonging to eleven species, namely *Aedes albopictus*, *Aedes caspius*, *Aedes detritus*, *Aedes dorsalis*, *Aedes rossicus*, *Aedes vexans*, *Anopheles maculipennis*, *Culex pipiens*, *Culex modestus*, *Culiseta annulata* and *Culiseta longiareolata.* The six BAM devices trapped 84,461 mosquitoes. The average capture rate per BAM is 76.92 mosquitoes per day. The rate of nuisance has decreased from 4.33 ± 2.88 before the deployment to 1.59 ± 2.77 after BAM implantation. The Qista BAM trap seems to be an excellent tool for reducing the nuisance rate and may help researchers to optimize trapping methods by obtaining more significant sample sizes. It may also allow the updating of the host-seeking mosquito species’ reported biodiversity in the south of France.

## 1. Introduction

The mosquito family (Diptera: Culicidae) includes over 3614 recognized species [1,2]. The importance of mosquitoes in human and animal health is primarily due to their ability to transmit agents of severe infectious diseases [3,4,5]. They are the primary vectors of several bacterial, parasitic, and arboviral diseases (malaria, dengue, Zika, chikungunya, yellow fever, Rift Valley fever, and West Nile) [6,7]. Malaria is one of the most critical worldwide mosquito-borne diseases, transmitted mainly by the *Anopheles* mosquito genera [8,9]. The estimated number of deaths in 2020 was 627,000 (1.2 deaths per minute), while the number of cases was 241 million (7.6 cases per second) [10]. With over 3.3 million cases prior to October 2022, dengue is the most prevalent *Aedes*-mosquito-borne disease, causing significant world public health concerns [11,12,13]. In 2022, France reported 227 imported cases of dengue and 65 autochthonous cases [14]. These statistics represent more than the total number of cases in the last ten years in France.

Mosquito vector control is considered the most effective way to prevent the spread of vector-borne diseases, as vaccines and/or efficient treatments are not always available [15,16]. Several strategies have been implemented to control mosquitoes and their role as vectors, including chemical, biological, physical, and integrated control methods [17,18,19]. Overall, utilizing a combination of control measures known as integrated vector management has turned out to be a powerful method for managing and reducing the spread of diseases transmitted by mosquitoes [18]. The chemical approach involves the application of insecticides, either targeting adult mosquitoes or their larvae [20]. Since the introduction of dichlorodiphenyltrichloroethane (DDT) in the 1940s, various products have been used to control mosquitoes and their ability to transmit diseases, making this chemical an effective primary intervention [21]. Unfortunately, this method has several limits and inconveniences, including affecting a wide range of insects, the development of resistance genes, and significant toxicity to animals and humans [22,23,24]. Insecticides, including pyrethroids, which are commonly used to control mosquitoes, have been found to have various negative effects on human health. These include impacts on fertility, immune and cardiovascular systems, hepatic metabolism, enzymatic activity, and the potential to induce nephro- and hepatotoxicity [23]. The biological method consists of the use of biological agents, such as bacteria (for example, *Bacillus thuringiensis israelensis (Bti*) or *Wolbachia*) [25,26], larvivorous fish and crustaceans [27,28], bats [29], and other biological fighters, including autocidal methods, such as the use of sterile insects (SIT) [4,30]. Recently, integrated vector management programs have increasingly been incorporating the use of SIT in mosquito control efforts. This method incurs significant expenses due to the costs associated with producing, sterilizing, and releasing the sterile insects. Additionally, failure resulting from inadequate preparation can have significant and long term negative consequences [30]. Likewise, *Bti* is the most commonly used as a biological method to control mosquitos. However, despite its biological nature, this strategy may be limited in its ability to control mosquitoes on a large scale and may have adverse effects on the environment [25,31]. The primary reason for this constraint is the persistent exposure of mosquito populations to accumulated *Bti* and its toxins, resulting in a gradual decline in their susceptibility over time. [25].

As an environmentally friendly alternative, physical methods of controlling mosquitoes may be promising strategies. This involves suppressing insects through physical or mechanical means [4,32]. Physical measures, including traps such as BGS (Biogents Sentinel), mosquito magnet, and various trapping strategies for adult mosquito removal, are commonly used in surveillance and integrated mosquito control programs [32,33,34]. Qista Techno BAM developed their initial prototype for mosquito traps in 2015 and was later patented under the application number WO/2016/020627 A1 [35] and tested in the field [36]. The Qista mosquito trap uses CO_2_ and olfactive lures to attract mosquitoes [36].

This study aims to investigate the effectiveness of an innovative mosquito trap system in reducing the abundance of adult mosquitoes and to evaluate the mosquito species diversity in the Camargue region during peak mosquito season.

## 2. Materials and Methods

### 2.1. Ethics Statement

The present field study did not require any specific authorization. It did not involve any endangered or protected species. The Qista Techno BAM has generously lent six Qista mosquito traps to Espeyran Castle as part of a skills sponsorship. The goal was to protect the area around the castle during the “Heritage Day” event and also to provide the authors with the opportunity to evaluate the trap’s efficiency and inventory the mosquito species in the Camargue regions in the subsequent period. Authorization to access the castle twice a week to collect mosquitoes in the BAMs and perform human landing catches (HLC) was obtained and included in the charter of commitment. The researchers who carried out the HLC and retrieved the mosquito nets from the traps were made fully aware of the potential risks, gave their consent, and remained in constant communication with local health authorities. The study was conducted in an area where mosquito-borne diseases had not been reported, and there were no reports of disease transmission by local authorities during the study’s duration.

### 2.2. Study Area

The study was conducted at Espeyran Castle (43.644404, 4.403153), located near the municipality of Saint-Gilles in the Gard department, part of the Camargue region, France. The castle was once part of the Sabatier d’Espeyran family estate, spanning 535 hectares and surrounded by vineyards extending over a radius of more than one kilometer, as well as being delimited by the “Rhône à Sète” canal. The castle is located three kilometers from the village of Saint-Gilles (Figure 1). It is currently used as the national center of microfilm and digitization and is attached to the national and territorial archives.

### 2.3. Description of the Trapping Methods

#### 2.3.1. The Qista BAM Trap

The Qista BAM trap (Figure 2) is made of a weather-resistant polypropylene box, measuring 101.5 cm in height, 52 cm in depth, and 32.3 cm in width. On the back side of the traps a large compartment is located in which a CO_2_ cylinder is placed. The CO_2_ cylinder is connected to a duct that leads to a diffusion chimney at the top of the compartment. Inside it, an octenol-based olfactive lure in the form of impregnated polymer beads is placed. In front of the CO_2_ cylinder, an inverted fan acts as a vacuum. On the front face, inside a small box, a 1 mm × 0.5 mm mesh net is connected to a duct to catch the attracted mosquitoes.

A connected smartphone application, “Qista App”, directly linked to a main board installed in the trap, can remotely control the operating program. A probe connected to the main board allows users to access hourly meteorological data, such as temperature, barometric pressure, and humidity. The app also allows users to monitor captures in real time through sensors installed in the trap.

Overall, the Qista BAM releases an octenol-based lure and carbon dioxide from the CO_2_ cylinder at 0.2 L/min to attract female mosquitoes to the trap entrance. Once near the entrance, a power-supplied impeller fan sucks the mosquitoes into a net.

#### 2.3.2. The Site-Nuisance Characterization

In order to deploy the Qista BAMs at the study site, it was determined that the Espeyran Castle was particularly concerned with diurnal mosquitoes (this corresponds to the castle daytime activity). The authors, all entomologists, had already gained insight into mosquito nuisances in Camargue. Two pre-survey visits were conducted at the site study on 12 May and 18 May 2021. The presence of marshes and vineyards bordering the castle, as well as the presence of different larval habitats within the castle’s spaces, were documented. Based on these observations, two pertinent areas were identified: an area to be protected—corresponding to the surroundings of the castle (the future location of the Heritage Day event)—and a second control area, located 130 m from the protected area. To ensure that the nuisance levels were homogeneous between the control and protected areas before deploying the traps, three locations in each area were identified as follows:S1, S2, and S3, located 30 m from the castle (Figure 3). These were considered to be located in the protected area, as the BAM traps have an attraction radius of up to 60 m [30].S4, S5, and S6, located 130 m from the castle and 70 m from the traps (Figure 3). These sites were considered to be located in the control area because they were located outside the attraction radius of the BAM traps, which is 60 m.

On 12 May and 18 May, HLC were conducted in the six locations (S1, S2, S3, S4, S5, S6), (Figure 3). During the HLC, an operator with exposed legs sat still for 15 min, and any mosquitoes that landed on them were caught using a mouth aspirator. All collected mosquitoes that landed on the operator’s leg were counted and identified at species level. The pre-implantation-nuisance levels were then compared between the two zones.

#### 2.3.3. The Implantation Study

The implantation study involved analyzing the environment to determine the appropriate number and locations for the BAM traps. Based on the pre-surveys of 12 and 18 May 2021, six BAM traps were deployed in a belt formation to intercept mosquitoes, with the following coordinates: BAM 1 (43°38′38.19″ N, 4°24′10.13″ E), BAM 2 (43°38′37.65″ N, 4°24′11.18″ E), BAM 3 (43°38′37.18″ N, 4°24′12.62″ E), BAM 4 (43°38′38.37″ N, 4°24′14.55″ E), BAM 5 (43°38′39.51″ N, 4°24′15.82″ E), and BAM 6 (43°38′40.82″ N, 4°24′16.20″ E) (Figure 3). An octenol-based lure called “moustique traditionnel” was used in all six traps throughout the experiment. The CO_2_ cylinders and lures were replaced every three weeks. During the 28-week experiment, 54 CO_2_ cylinders and 54 olfactive lure boxes were used.

#### 2.3.4. Experiment Design: Monitoring the Qista BAM Trap Potential through HLC

At the outset of the study, it was deemed necessary to conduct two sampling rhythms concerning mosquito net retrieval, specifically a 24-hour and a 6-day capture period. Mosquitoes captured within a 24 h period were often of excellent preservation quality, allowing easy morphological identification. The 24 h sampling frequency also provides an excellent quantitative indicator of BAM performance over a 24 h capture period. However, with regard to the 6-day capture period, we sought to obtain all the captures of the season. Therefore, we supplemented the 24 h captures with those of the 6-day captures and estimated that this capture frequency would provide qualitative results of a maximum number of species captured over the entire season. The experiment aimed mainly to target *Aedes caspius* and *Ae. detritus* mosquitoes, responsible for causing a nuisance during the castle’s opening hours, and these have also been the targets of *Bti* treatment in the Camargue area as a whole. The study was conducted from 18 May to 17 November. On 18 May 2021, the six BAM traps were deployed at the designed locations. Two different interventions were conducted. The first corresponds to a time frame of Tuesday to Wednesday, in which mosquitoes were captured over 24 h periods. The second intervention corresponds to a time frame from Wednesday to the following Tuesday of the next week; in this case, mosquito captures occurred over a period of 6 days. The BAM nets were recovered and immediately replaced with empty nets after each intervention. The effectiveness of the mosquito traps was evaluated not just by considering the number of mosquitoes captured in the BAM nets but also by their ability to reduce the number of mosquito bites on humans. HLCs were carried out on Tuesday and Wednesday at fixed times between 10 a.m. and 12 p.m. at all the six sites in order to compare the daytime nuisance between the protected and the control areas.

Once in the laboratory, all mosquito captured via both the trap and HLC were individually identified at the species level using the Xper2 dichotomic keys [37]. If the mosquitoes were still alive, they were put into a freezer for 3–5 min. Once frozen, the mosquitoes were placed in a Petri dish under a stereomicroscope before being identified. Throughout the 28 weeks of the experiment, more than 250 h were invested in the field (≈4–5 h/intervention day) and around 160 h were spent in the laboratory (≈3 h/day).

### 2.4. Data Analysis

All analyses were performed using Rstudio version 4.1.2 (01 November 2021), “Bird Hippie” Copyright (C) 2021, the R Foundation for Statistical Computing.

## 3. Results

### 3.1. Inventory of the Captured Mosquitoes

A total of 85,600 mosquitoes were captured using the six Qista BAMs (98.67% (84,461/85,600)) and the HLC (1.33% (1139/85,600)). Overall, eleven mosquito species were caught at Espeyran Castle, namely *Aedes albopictus*, *Aedes caspius*, *Aedes detritus*, *Aedes dorsalis*, *Aedes rossicus*, *Aedes vexans*, *Anopheles maculipennis*, *Culex pipiens*, *Culex modestus*, *Culiseta annulata*, and *Culiseta longiareolata* (Figure 4A). The six weeks between 8 September and 19 October represent the highest capture period of the survey, during which 58.96% (50,470/85,600) mosquitoes were caught. *Ae. caspius* was the only species caught at least once throughout the study period via one or more trapping methods, with a peak capture rate between 8 September and 19 October (Figure 4B).

### 3.2. HLC before Implantation

On 12 May and 18 May, two HLCs were performed to estimate the nuisance at the six sites (three sites in the protected areas and the others in the control areas) before the deployment of the Qista traps. The average capture value using HLC was 4.33 ± 2.88 in the protected regions versus 4.17 ± 3.37 in the control areas (Figure 5). The Wilcoxon test (W = 21, *p*-value = 0.66 > 0.05) indicated no significant proven difference in median catches between the protected and control areas before the BAM implantation. Additionally, no statistically significant difference was observed among the six positions of the sites before the BAM’s implantation (Kruskal–Wallis test, χ^2^ = 10.821, df = 5, *p*-value = 0.055 > 0.05).

### 3.3. The Qista BAM Catches

A total of 84,461 mosquitoes were caught using the six Qista BAMs at Espeyran Castle from 18 May to 17 November. During this six-month-long field study, the average number of captures per BAM per day was 76.92 mosquitoes. Between 8 September and 19 October, we achieved the highest number of captures during the period of study, where all six BAM devices caught 59.03% (49,860/84,461) of the total mosquitoes (Figure 6A). During this study, eleven mosquito species were captured by the BAM, including *Ae. albopictus* 0.05% (43/84,461), *Ae. caspius* 62.47% (52,766/84,461), *Ae. detritus* 8.19% (6913/84,461), *Ae. dorsalis* 0.1% (87/84,461), *Ae. rossicus* 0.02% (16/84,461), *Ae. vexans* 24.28% (20,506/84,461), *An. maculipennis* 0.94% (795/84,461), *Cx. pipiens* (3.51%, 2967/84,461), *Cx. modestus* 0.01% (7/84,461), *Cs. annulata* 0.42% (56/84,461), and *Cs. longiareolata* 0.01% (5/84,461) (Figure 6B). BAM6 caught 45.66% (38,566/84,461) of the total mosquitoes vs. 9.69% with BAM1 (8188/84,461), 9.38% (7926/84,461) with BAM2, 14.16% (11,963/84,461) with BAM3, 17.58% (14,856/84,461) with BAM4, and 3.5% (2962/84,461) with BAM5.

#### 3.3.1. The 24 h Net Recovery

During the 27 sampling days, 19,726 mosquitoes were caught using the Qista-BAM devices. The average capture rate of the recovery net during 24 h per BAM and per day was 121.76 ± 129.90 mosquitoes. BAM6 alone reached a record of 1205 captures in one day (day 20). The period between day 17 to day 22 (five capture days from 8 September to 12 October) represented the highest capture period of the study, where the six BAMs caught 60.66% (11,966/19,726) of the total mosquitoes. All BAMs recorded a maximum capture on day 22, with 2873 mosquitoes captured. This represents 14.56% (2873/19,726) of all the mosquitos caught in 24 h by the BAMs. Except for *Cs. longiareolata*, all eleven mosquito species mentioned above were identified in the 24 h captures. *Ae. caspius* was the most prevalently caught species, with 67.78% (13,371/19,726) (Figure 7A). BAM6 caught 41.18% (8123/19,726) of the total mosquitoes vs. 9.52% with BAM1 (1878/19,726), 8.85% (1747/19,726) with BAM2, 16.75% (3305/19,726) with BAM3, 19.39% (3825/19,726) with BAM4, and 4.3% (848/19,726) with BAM5 (Figure 7B).

#### 3.3.2. The Six-Day Net Recovery

Throughout the 26 sampling weeks, 64,735 mosquitoes were caught using the BAMs. The average of captures per BAM every six days was 414.97 ± 434.80 mosquitoes, corresponding to 69.16 ± 72.47 mosquitoes per BAM per day. BAM6 alone achieved a record of 5863 captures in one day of capture during week 18. The period between week 17 to week 21 represents the highest capture period of the study, where all the BAMs caught 56.03% (36,277/64,735) of the total mosquitoes. The six BAMs recorded maximum captures during week 20, with 9003 mosquitoes captured (Figure 8A). This represents 13.90% (9003/64,735) of the total mosquitos caught in 6 days by the BAMs.

All the eleven mosquito species previously mentioned were identified (Figure 8A), with *Ae. caspius* (39,395/64,735) representing 67.78% of the total mosquito species captured. BAM1, 2, 3, 4, 5, and 6 caught, respectively, 9.74% (6310/64,735), 9.54% (6179/64,735), 13.37% (8658/64,735), 17.04% (11,031/64,735), 3.26% (2114/64,735), and 47.02% (30,443/64,735) of the total mosquitoes.

#### 3.3.3. Six-Day vs. Twenty-Four-Hour Net Recovery 

In order to provide a comprehensive evaluation of the BAM performance, data on the frequency, total number of mosquitoes, and abundance of each of the eleven species were analyzed for the 24-hour and 6-day catches (Appendix A). Through the utilization of statistical tests, the comparisons of sums, percentages, means, and medians of the species caught were obtained.

Significant differences in catch frequencies were observed among several mosquito species, including *Ae. caspius*, *Ae. detritus*, *Ae. vexans*, *Ae. rossicus*, *An. maculipennis*, *Cs. annulata*, and *Cx. pipiens*. Notably, *Ae. caspius*, *Ae. Rossicus*, and *Cs. annulata* exhibited higher catch frequencies in the 24 h capture periods compared to the 6-day catch periods. Conversely, *Ae. detritus*, *Ae. vexans*, *An. maculipennis*, and *Cx. pipiens* revealed higher catch frequencies in the 6-day catch periods as opposed to the 24 h capture periods. Moreover, no significant differences were noted concerning *Ae. dorsalis*, *Ae. albopictus*, *Cs. longiareolata*, and *Cx. modestus* regarding the two types of catches (Table 1).

The comparison of medians between the total mosquito captures and species in the 24-hour vs. 6-day catch periods revealed significant differences, as indicated by the *p*-values from the Wilcoxon test for *Ae. caspius*, *Ae. detritus*, *Ae. vexans*, *Cx. pipiens*, and the overall sum of captured mosquitos. In terms of the mean of captures, those of the 6-day capture periods were found to be higher than those of the 24 h capture periods for the cited species and for the overall catches. However, no differences were noted concerning *Ae. dorsalis*, *Ae. rossicus*, *Ae. albopictus*, *An. maculipennis*, *Cs. annulata*, *Cs. longiareolata*, and *Cx. modestus* during either the 6-day or 24-hour capture periods (Table 1).

### 3.4. HLC in the Control Area vs. the Protected Area after the BAM Deployment

After the deployment of the BAMs on 18 May, 53 HLC samplings were carried out at each of the six sites, resulting in a total of 318 samplings in both the protected and control areas combined. A total of 861 and 227 mosquito specimens were collected, respectively, in the control and protected areas. This corresponds to an average of 5.41 ± 9.51 in the control areas and 1.42 ± 2.77 in the protected areas after BAM implantation (Figure 9A). According to the Kruskal–Wallis test, between the protected areas’ three sites and the control areas’ sites, there was a significant difference in capture rates between the two areas (χ^2^ = 48.429, df = 5, *p*-value = 2.903 × 10^−9^). Five species were identified in the protected areas, namely *Ae. albopictus* 0.40% (1/253), *Ae. caspius* 90.51% (229/253), *Ae. detritus* 3.56% (9/253), *Ae. dorsalis* 0.40% (1/253), and *Ae. vexans* 5.13% (13/253). In contrast, six mosquito species were caught in the control areas: *Ae. albopictus* 0.23% (2/886), *Ae. caspius* 85.10% (754/886), *Ae. detritus* 2.14% (19/886), *Ae. dorsalis* 1.47% (13/886), *Ae. vexans* 10.95% (97/886), and *Cx. pipiens* 0.11% (1/886). The pairwise comparison between the control site before and after implementation indicated no significant difference (Kruskal–Wallis; χ^2^ = 2.579, df = 5, *p*-value = 0.764). These results have proven that the nuisance was unchanged between the study’s beginning and its end (Figure 9B). However, the *p* value of the protected site before and after implementation showed a highly significant difference (Kruskal–Wallis; χ^2^ = 11.386, df = 5, *p*-value = 0.044) (Figure 9C).

## 4. Discussion

This study provides valuable data on mosquito species’ inventory in the Camargue region (south of France). Overall, eleven mosquito species were caught using HLC and BAM Qista traps, namely *Ae. albopictus*, *Ae. caspius*, *Ae. detritus*, *Ae. dorsalis*, *Ae. rossicus*, *Ae. vexans*, *An. maculipennis*, *Cx. pipiens*, *Cx. modestus*, *Cs. annulata*, and *Cs. longiareolata*. The performance of the Qista BAM barrier trap system was evaluated weekly. Six BAMs were deployed at Espeyran Castle near Saint-Gilles. The study was based on evaluating the nuisance reduction between the treated area (protected) and a control area using the HLC method. Overall, the traps recorded up to 1205 mosquitoes during a single operating day. All 84,461 BAM-captured mosquitoes, whether captured over a period of 6 days or 24 h, were well preserved and easily morphologically identifiable. Consequently, this indicates that the traps do not damage the captured mosquitoes and can preserve their morphological integrity for up to 6 days inside the net trap. Eleven mosquito species within four genera (*Aedes*, *Culex*, *Anopheles*, and *Culiseta*) were captured by the BAMs. The timing of net retrieval was found to have a significant impact on the frequency, abundance, and number of species caught in terms of *Ae. caspius*, *Ae. detritus*, *Ae. vexans*, and *Cx. pipiens* over a 24-hour or 6-day period. However, no significant impact was observed for *Ae. dorsalis*, *Ae. albopictus*, *Cs. longiareolata*, and *Cx. modestus*. Additionally, it was found that the timing of net retrieval had an impact only on the frequencies of *Ae. rossicus*, *An. maculipennis*, and *Cs. annulata*.

*Aedes caspius*, *Aedes vexans*, and *Aedes detritus* were the dominant species captured in the traps. These species are known to be abundant in the Camargue region, where they bite humans, horses, and birds [37]. *Aedes caspius* has been proven to transmit the West Nile virus in the laboratory [38] and in field conditions in Italy [39]. *Aedes vexans* collected in the field in Germany [4,17] and the United States [40,41] have been found positive for West Nile virus. In the laboratory, they were found to transmit the chikungunya [42], Zika [43], and West Nile viruses [44]. In the UK, *Aedes detritus* mosquitoes were positively evaluated to transmit dengue and chikungunya in laboratory experiments [45]. In Italy, they were also found positive for the Usutu virus in the field [46]. On another hand, *Culex pipiens* was among the predominantly caught mosquito species by the BAM. *Culex* species are recognized as causing significant discomfort and as carriers of viruses such as Zika, Sindbis virus, Usutu, and West Nile virus in Europe [46,47]. Concerning the *Anopheles* genus, *An. maculipennis* was the fifth most caught mosquito species by the BAM, but none were captured by HLC. This can be explained by the fact that *An. maculipennis* is a nocturnal mosquito and the HLC was performed in diurnal periods. Although *An. maculipennis* is a native European mosquito, it can be considered receptive to malaria parasites. However, this receptivity is of minor importance [48]. The capture of the tiger mosquito, *Ae. albopictus* by the BAMs at the study site caught our attention. Indeed, *Ae. albopictus* is known to exhibit aggressive diurnal biting [49]. Of note, only 1/227 mosquitoes in the protected area and 2/861 *Ae. albopictus* in the control area were caught by HLC vs. 43/84,461 captures by the BAMs. Despite the few numbers of *Ae. albopictus* specimens present in the study area, BAMs may reduce human contact with *Ae. albopictus* and could be used as an excellent method of trapping and monitoring the tiger mosquito. As a reminder, *Ae. albopictus*, an established vector of Zika, dengue, chikungunya, and other vectorized diseases, has been present in France since 2004 [50,51]. These *Aedes*-borne diseases are particularly endemic in the south of France [52,53]. Finally, five other species were anecdotally captured by the BAMs: *Ae. dorsalis*, *Ae. rossicus*, *Cx. modestus*, *Cs. annulata*, and *Cs. longiareolata*, which may confirm the efficiency of this device as a mosquito trap.

During the study period, the biting rate was assessed before and after BAM deployment and was found to be evenly distributed across the control and protected areas. Although the assessment was conducted throughout a single time period, a substantial decrease in the nuisance level was observed in the protected area throughout the study period compared to before BAM implantation. Assessing the efficacy of mosquito traps can be carried out using different methods. When evaluating the effectiveness of various traps in the same experiment in the field, the Latin square is the optimal technique for obtaining a comparative view on mosquito nuisance reduction [54]. This type of design reduces the effects of confounding variables and obtains more accurate results. However, when evaluating one trap, although the number of trapped mosquitoes may serve as a relative indicator for trap performance, the primary metric for evaluating their overall effectiveness should be the decrease in the number of mosquito bites on humans [55].

HLC can be the main useful indicator to assess effectiveness, as it provides a direct and valuable measurement of the nuisance of mosquito bites. Englbrecht et al. evaluated the effectiveness of BG-Sentinel traps in controlling *Ae. albopictus* populations in Italy, where three intervention sites were compared to three similar control sites. The intervention sites, which were equipped with BG-Sentinel traps placed outdoors, showed an average reduction of 87% of *Ae. albopictus* collected by HLC over the entire study period [56]. In southern France, Akhoundi et al. assessed the “Biobelt anti-Mosquitoes” efficacy for controlling exophilic adult *Aedes* populations using the HLC technique [34].

Overall, this study showed that the Qista BAM mosquito trap, which uses CO_2_ and olfactive lures as bait, performed well in catching outdoor host-seeking mosquitoes. If used in combination with other control measures, such as reducing larval breeding sites, the BAM has the potential to be an effective tool for controlling mosquitoes. More research is needed to examine the BAM’s long-term effectiveness.

## 5. Conclusions

Mosquitoes are known to cause various inconveniences, including acting as vectors for deadly diseases. To control mosquitoes and vector-borne diseases, various strategies are being implemented, including the use of Qista traps. In this study, we identified eleven mosquito species in the Camargue region, including *Aedes albopictus*, the main vector of dengue in Europe, which posed an unusual challenge for France in 2022. Despite the low presence of *Ae. albopictus* in the study area, they were successfully captured by the BAMs. The main mosquitoes responsible for daytime nuisance at Espeyran Castle, namely *Aedes caspius*, *Aedes detritus*, and *Aedes vexans*, were well-represented in the catches. Regarding the nocturnal mosquito, the BAMs trapped mostly *Culex pipiens* and *Anopheles maculipennis*. The traps were able to catch a large number of mosquitoes samples. They were also found to be effective in reducing the nuisance rate caused by mosquitoes. This was determined through a comparison of the treated site and the control site using the human landing catches method. The study demonstrated the good performance of the Qista BAM mosquito-baiting, human-odor, CO_2_-baited trap for outdoor host-seeking mosquitoes in Saint-Gilles, Camargue, France. Combined with other control methods, such as suppressing larval breeding sites, the BAM could be a promising mosquito-controlling tool. Further studies should be conducted to investigate the dynamic recolonization of the studied area, allowing us to gain a better view of long-term efficacy.

## Figures and Tables

**Figure 1 animals-13-01809-f001:**
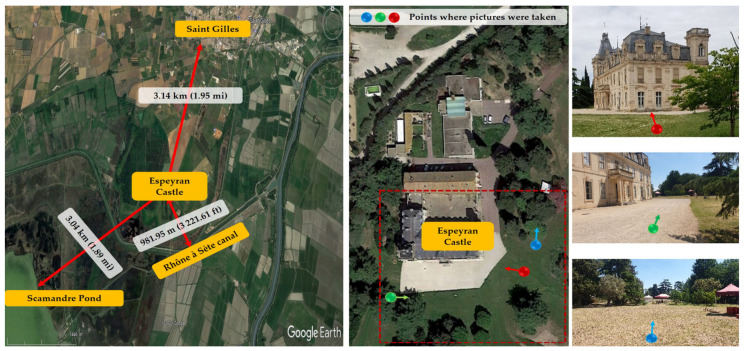
Location of the Espeyran Castle in Saint Gilles, France.

**Figure 2 animals-13-01809-f002:**
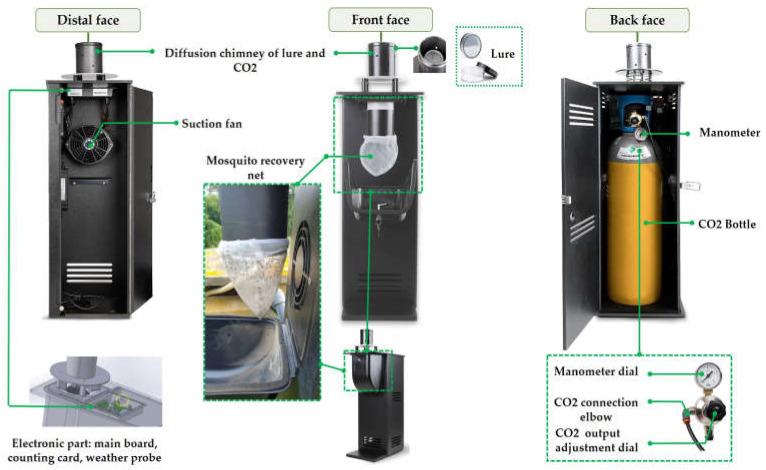
The QISTA BAM trap composition.

**Figure 3 animals-13-01809-f003:**
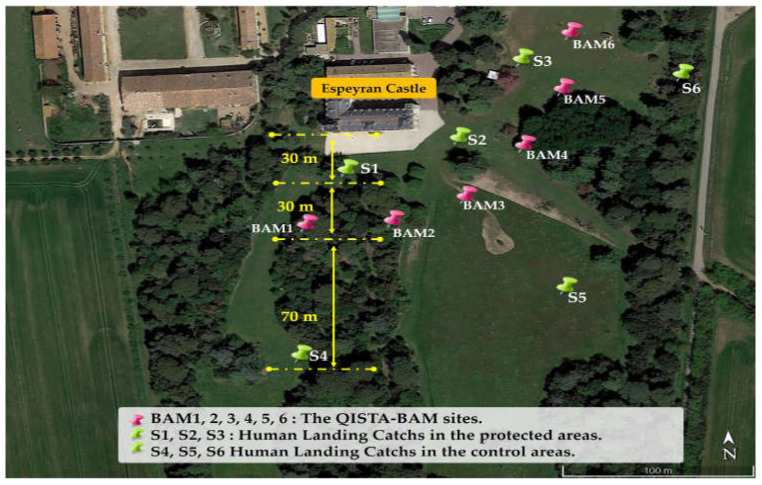
The sites of the experiment.

**Figure 4 animals-13-01809-f004:**
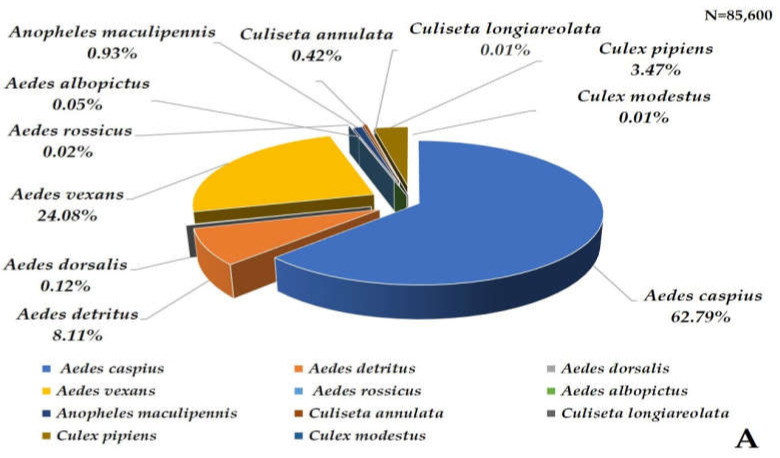
Mosquito captures at Espeyran Castle. (**A**) mosquito species inventory at Espeyran Castle. (**B**) dynamics of the captures (via BAM or HLC) during the study period.

**Figure 5 animals-13-01809-f005:**
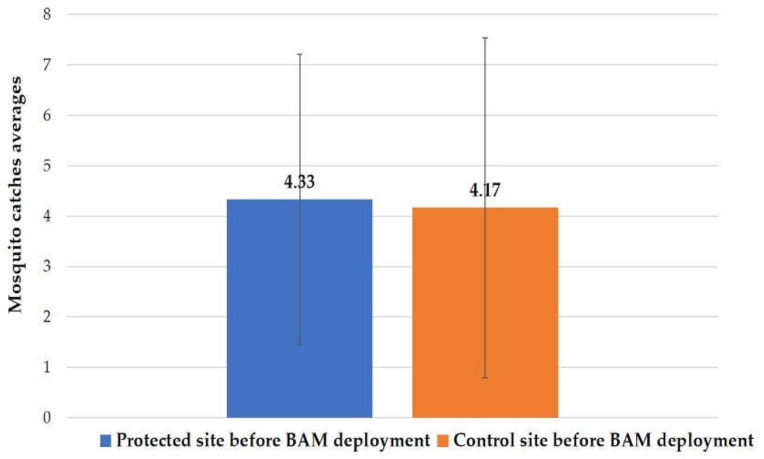
Mosquito catch averages via HLC before BAM deployment in the protected and control areas.

**Figure 6 animals-13-01809-f006:**
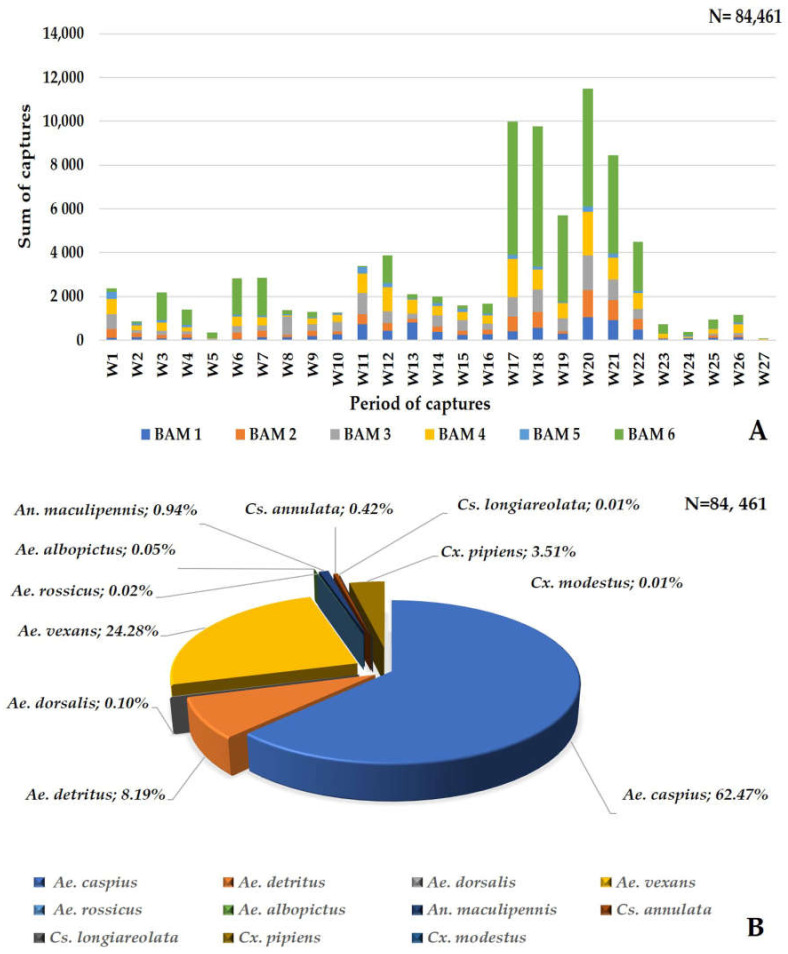
Mosquito captures by the six BAMs (**A**) dynamics of the captures by the BAMs during the study period. (**B**) mosquito species captured by the BAMs.

**Figure 7 animals-13-01809-f007:**
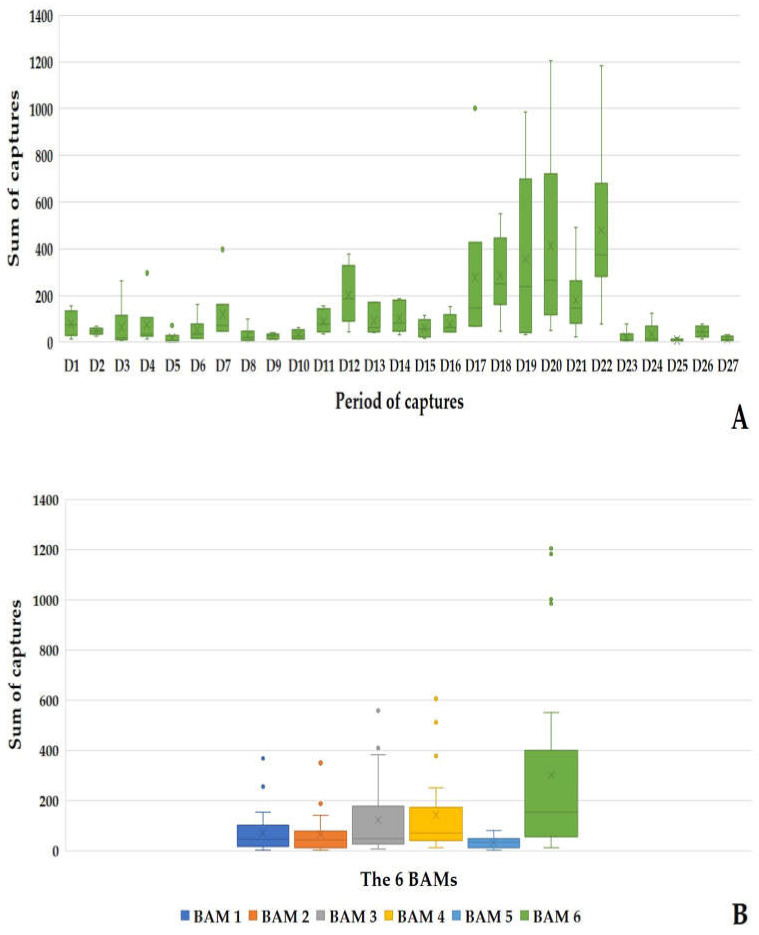
Results of the 24 h mosquito captures by the BAMs. (**A**) dynamic of the mosquito captures by the BAMs during the 24 h captures. (**B**) Sum of the mosquito captured by each BAM during the 24 h captures.

**Figure 8 animals-13-01809-f008:**
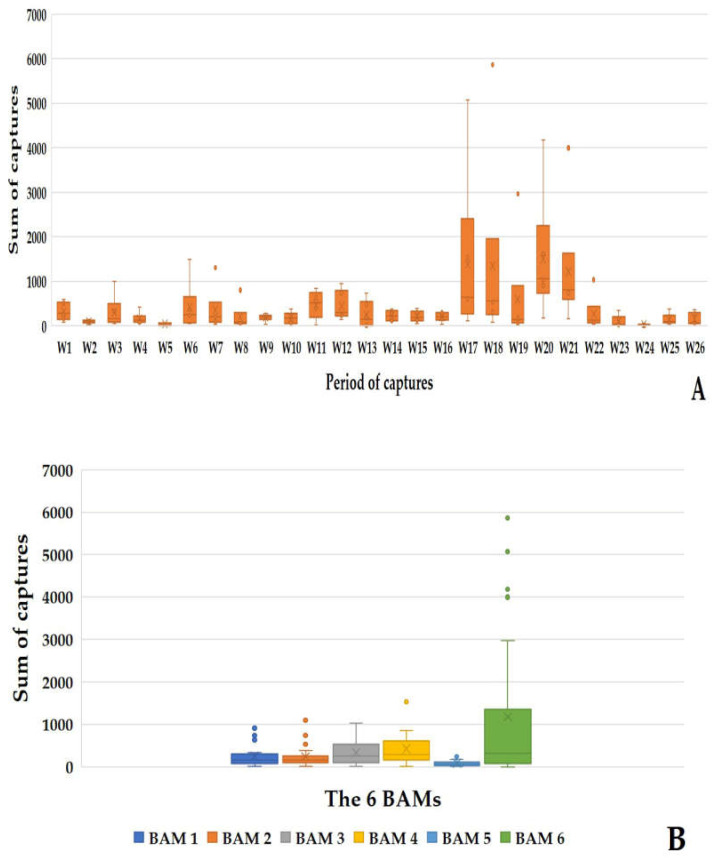
Results of the mosquito captures by six BAMs during the six-day capture period. (**A**) Sum of the mosquito captures by the BAMs during the 6-day capture period. (**B**) Sum of mosquito captures by each BAM during the 6-day capture period.

**Figure 9 animals-13-01809-f009:**
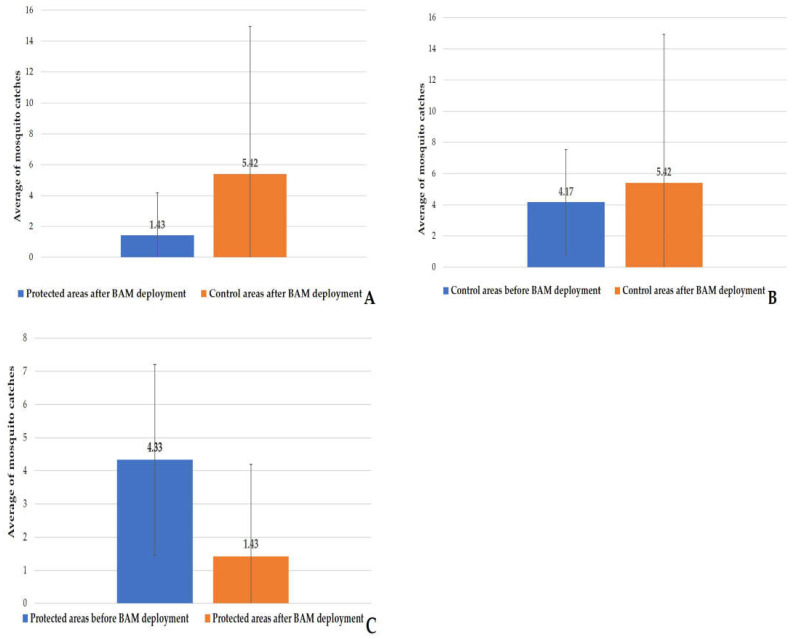
Human landing catches after BAM deployment. (**A**) HLC after BAM deployment in the protected and control areas. (**B**) comparison of HLC before and after BAM deployment in the control areas. (**C**) comparison of HLC before and after BAM deployment in the protected areas.

**Table 1 animals-13-01809-t001:** Comparison of the number of captured mosquitoes, abundances, and frequencies of each species between the 24-hour and 6-day capture periods.

Species	∑ of the 24 h Capture Periods	∑ of the 6 d Capture Periods	χ^2^ between 24 h and 6 d Capture Periods	Mean of 24 h Capture Period/BAM	Mean of 6 d Capture Period/BAM	*p*-Value 24 h vs. 6 d Catches
*Ae. caspius*	13,371 (67.78%)	39,395 (60.86%)	χ^2^ = 309.22, df = 1, *p*-value < 2.2 × 10^−16^ *	82.54 ± 83.53	252.53 ± 275.50	0.00049 *
*Ae. detritus*	1150 (5.83%)	5763 (8.90%)	χ^2^ = 189.53, df = 1, *p* -value < 2.2 × 10^−16^ *	7.10 ± 13.27	36.94 ± 44.43	0.0072 *
*Ae. dorsalis*	24 (0.12%)	63 (0.10%)	χ^2^ = 0.65042, df = 1, *p* -value = 0.42	0.15 ± 0.41	0.40 ± 1.16	0.87
*Ae. vexans*	4494 (22.78%)	16,012 (24.73%)	χ^2^ = 31.248, df = 1, *p* -value = 2.271 × 10^−8^ *	27.74 ± 56.26	102.64 ± 170.58	0.036 *
*Ae. rossicus*	12 (0.06%)	4 (0.01%)	χ^2^ = 21.046, df = 1, *p* -value = 4.483 × 10^−6^ *	0.07 ± 0.25	0.03 ± 0.13	0.33
*Ae. albopictus*	9 (0.05%)	34 (0.05%)	χ^2^ = 0.038286, df = 1, *p* -value = 0.8449	0.06 ± 0.11	0.22 ± 0.55	0.43
*An. maculipennis*	145 (0.74%)	650 (1.00%)	χ^2^ = 11.449, df = 1, *p* -value = 0.0007155 *	0.90 ± 1.39	4.17 ± 6.62	0.066
*Cs. annulata*	111 (0.56%)	245 (0.38%)	χ^2^ = 41.059, df = 1, *p* -value = 1.477 × 10^−10^ *	0.69 ± 1.75	1.57 ± 4.72	0.49
*Cs. longiareolata*	0 (0.00%)	5 (0.01%)	χ^2^ = 0.49823, df = 1, *p* -value = 0.4803	0	0.03 ± 0.11	0.076
*Cx. pipiens*	407 (2.06%)	2560 (3.95%)	χ^2^ = 159, df = 1, *p* -value < 2.2 × 10^−16^ *	2.51 ± 2.74	16.41 ± 14.69	0.00010 *
*Cx. modestus*	3 (0.02%)	4 (0.01%)	χ^2^= 0.59737, df = 1, *p* -value = 0.4396	0.02 ± 0.10	0.03 ± 0.13	0.98
Total mosquito species	19,726	64,735	NA	121.765 ± 129.90	414.97 ± 434.80	8.6 × 10^−5^ *

* = significant differences.

## Data Availability

All relevant data are within the paper and its Appendix A.

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
