# Peer review of "Assessing the Effectiveness of Qista Baited Traps in Capturing Mosquito Vectors of Diseases in the Camargue Region (France) and Investigating Their Diversity"

_animals, 2023, doi:10.3390/ani13111809_

Round 1
Reviewer 1 Report
Results of the manuscript are interesting and moderately well redacted. The manuscript can be accepted with intermediate revision. The description of the objectives and methodology need to be improved to clarify the point of the study. Results also need to be lightly restructured to improve the reader understanding.
The main problem is that it is difficult to unravel the three different outcomes/indicators collected by the BAM traps and why the authors chose to take them into account.
Also, authors do not make very clear the HLC design in the methodology and they mingle the HLC and BAM traps results in the first part of the results. Since each capture method has a particular role in the study, it is not accurate to do so. They are two different kinds of results.
About the figures, I think that authors must pay more attention to their quality: no border, readable letters, all the data, etc.
I am not English native, but it seems that the redaction must be corrected, and authors must pay attention to some important details of the text.
All the best.

Author Response
We would like to extend our appreciation to Reviewer 1 for their valuable contribution. We have thoroughly addressed all the recommended changes, and a response document detailing our actions has been enclosed.

Reviewer 2 Report
Overall, this is an interesting manuscript. I like the comparison between traps and landing rates. One suggestion that I have is that the authors send this manuscript to a native English speaker for editing. There are some sentences that are awkward and at times stilted. I understand everything that they are trying to say, but it can be done much better. For example, this sentence is lacking something: "The castle was once part of 535 hectares family domain, surrounded by: vineyards 100 fields covering more than a one-kilometre radius and the Rhône à Sète Canal." I understand what it is conveying but the grammar is not exactly correct. There are many instances like this throughout the manuscript.
Make sure that all abbreviations are defined in the text at first use, not just in the abstract or summary.
Line 174: Repertory of species is not a phrase that is used often in English. Species inventory might be better.
Lines 175-181: This long litany would be better presented in a table. Same thing for lines 200 -208. Additionally, "mosquitos species" is incorrect. Eleven mosquito species (agreement of adjective and noun) is correct.
Line 247: 0.05 instead of 0,05.
Line 249: Generally "between" is used for comparisons of two things whereas "among" is used for three or more things.
Line 258: How did the authors perform a day? Reword this.
Author Response
We express our gratitude to Reviewer 2 for the valuable input. The remarks were thoroughly considered, and we have implemented all the necessary changes accordingly.
R2 : Make sure that all abbreviations are defined in the text at first use, not just in the abstract or summary.
Author answer: Okay done
Line 174: Repertory of species is not a phrase that is used often in English. Species inventory might be better.
Author answer: Okay done
Lines 175-181: This long litany would be better presented in a table. Same thing for lines 200 -208. Additionally, "mosquitos species" is incorrect. Eleven mosquito species (agreement of adjective and noun) is correct.
Author answer: Okay done
Line 247: 0.05 instead of 0,05.
Author answer: Okay done
Line 249: Generally "between" is used for comparisons of two things whereas "among" is used for three or more things.
Author answer: Okay done
Line 258: How did the authors perform a day? Reword this.
Author answer: Okay done

Round 2
Reviewer 1 Report
The manuscript is improved, but I recommend that some details should be attended before the paper can be published: the structure of the text should be improved and become more coherent, the figures have too long legends and the texts is not always readable. Also, the added texts should be english-reviewed.
The manuscript can be improved whether authors decide to explicitely make the comparison between 24hrs and 6 d BAM collects.
Please see the pdf manuscript with my specific comments.

Author Response
Response to reviewer comments:
Page and line numbers correspond to the “tracked” version of the manuscript, page and line number variations could occur between the “tracked” version and final version attributed to final file formatting.
Reviewer 1:
The manuscript is improved, but I recommend that some details should be attended before the paper can be published: the structure of the text should be improved and become more coherent, the figures have too long legends and the texts is not always readable. Also, the added texts should be english-reviewed.
The manuscript can be improved whether authors decide to explicitely make the comparison between 24hrs and 6 d BAM collects.
Please see the pdf manuscript with my specific comments.
R1, line 82 "costly... costly" repetitive, please change the redaction.
Author answer : changes done line 80 : This method incurs significant expenses due to the costs associated with ……..
R1, lines 87-89 : A better redaction is needed here. ei: "The continuous exposure of mosquito populations to Bti accumulation and toxin persistence are the main factors that lead them to a decrease in their susceptibility over time."
Author answer : changes done lines 86-88: The primary reason for this constraint is the persistent exposure of mosquito populations to accumulated Bti and its toxins, resulting in a gradual decline in their susceptibility over time.
R1, line 161 : Explain why 130 m is an important measure.
Author answer : changes done, as recommended by reviewer in another comment, we moved the section which talk about the six sites and explain why 130 m. lines 159-166.
R1, line 163 : Please indicate the belonging of the locations, ie: "... S1, S2 and S3 in the protected zone and S4, S5 and S6 in the control zone... "
Author answer : Done as recommended. details are in the added moved section lines 159-166.
R1, lines 189-155 : This section needs to be explained before.
Author answer : done as recommended. The section was moved up in the recommended place lines 159-166.
R1 lines 199-204 : The experiment with BAM traps was conducted from May 18th to November 17th. It was composed by two different interventions. The first one corresponds to the time frame from Tuesday to Wednesday, in which mosquitoes were captured over a 24-hour period. The second internvention corresponds to a the time frame from Wednesday to the following Tuesday of the next week, in this case, mosquito captures occured over a period of 6 days. The BAM's nets were recorevered and replaced with empy nets immediately after each intervention.
Author answer : section added as recommended. we thank the reviewer to improve the quality of the paper lines 208-212.
R1 lines 204-213 : The justification needs to be located at the beginning of this parragraph. After that authors explain the method.
Author answer : section moved at the beguining as recommended lines 187-195.
R1 line 207 :...often showed an...
Author answer : section moved as recommended in the previous comment, we added often showed an as recommended line 189.
R1 line 232 : …es.
Author answer : okay, done line 240, thank you to improve the quality of the paper.
R1 line 258 : Wilcoxon is a non-parametric test, so it tests median, not average. Authors should check with satistician whether they can talk about averages for a non parametric test.
Author answer : Thank you for your valuable feedback. You are right in pointing out that the Wilcoxon test is a non-parametric test that primarily focuses on the median rather than the average. We appreciate your suggestion, and make the necessary modifications to the text, however, the figures image the average and not the Wilcoxon median. Changes done line 260.
R1 line 264 : Correct the legend in accordance with the question of the average or median, ie: "Mosquito catches median or average by HLC before the BAM deployment in the protected and control zones."
Author answer : line 264 : the figure is about the average but the wilcoxon test compare the median. changes done line 266.
R1 line 285 : BAM
Author answer : done. Line 285
R1 line 287 As a recommendation: Why do not authors statistically test the number of captured mosquitoes, number of species, and frequencies of each species between the two different BAM interventions (24hr and 6 days)? They could report a table of such results, allowing a more easier way to compare the data. So, authors could integrate both section : 24 hrs and 6 days in one.
Author answer : excellent idea. We thank the reviewer to his insightful comments, we make the necessary adjustments accordingly lines 328-352 and in the section part lines 395-400.
R1 line line 344 control
Author answer : done control instead of unprotected (in all the paper).
R1 line 345 Eight hundred and sixty one and 227 mosquito specimens.
Author answer : okay done line 357.
R1 line 361 : Authors should review all the statisticall tests since I saw changes between the two versions of the manuscript.
Author answer : done, we have reviewed all the statisticall tests in the first round. When we have mentionned to implement the (df) in each Kruskal wallis results line 370-373.
R1 line 361 In a general way authors must review legends of all the figures to be informative and the most concise possible, to obey to the journal normativity, and to be congruent with the rest of the text (authors have many ways to refer to tested areas : protected/treated area/zone, unprotected/control area/zone, only chose one to be more understanble).
ei: "Figure 9. Mosquito catches average by HLC. A: in the protected and control zones after the BAM deployment. B: comparison before and after the BAM deployment in the control zone. C: comparison before and after the BAM deployment in the protected zone. "
Letters of all the figures must be readable with title of the vertical and horizontal axes.
Inside the figures, only put the letters A, B or C, but not Fig 9 A, Fig 9 B, Fig 9 C. Authors must check it for all the figures.
Author answer : okay dones, we reviewed accordingly all the figures.
R1 line 419 : Why talking about Latin square collection design? It is not evident taht authors used it to do their sampling. So authors should introduce the reason for talking about this design.
Second, Authors should be more consice in this section (about latin square)
Author answer : okay done , we resume the section to (When evaluating the effectiveness of different various traps in the same experiment in the field, the Latin square is the best optimal technique to have for obtaining a comparative view of on the mosquito nuisance reducing reduction among several traps [54]) lines 436-439.
